# Increased Risk of Gastric Cancer in Asbestos-Exposed Workers: A Retrospective Cohort Study Based on Taiwan Cancer Registry 1980–2015

**DOI:** 10.3390/ijerph18147521

**Published:** 2021-07-15

**Authors:** Yi-Jen Fang, Hung-Yi Chuang, Chih-Hong Pan, Yu-Yin Chang, Yawen Cheng, Lukas Jyuhn-Hsiarn Lee, Jung-Der Wang

**Affiliations:** 1Ph.D. Program in Environmental and Occupational Medicine, College of Medicine, Kaohsiung Medical University and National Health Research Institutes, Kaohsiung 807, Taiwan; fang531109@gmail.com (Y.-J.F.); ericch@kmu.edu.tw (H.-Y.C.); 2Digestive Disease Center, Show-Chwan Memorial Hospital, Changhua 500, Taiwan; 3Research Center for Environmental Medicine, Kaohsiung Medical University, Kaohsiung 807, Taiwan; 4Department of Environmental and Occupational Medicine, Kaohsiung Medical University Hospital, Kaohsiung 807, Taiwan; 5Institute of Labor, Occupational Safety and Health, Ministry of Labor, New Taipei City 221, Taiwan; chpan@mail.ilosh.gov.tw; 6National Institute of Environmental Health Sciences, National Health Research Institutes, Miaoli County 350, Taiwan; yuyin.chang@gmail.com; 7Institute of Health Policy and Management, College of Public Health, National Taiwan University, Taipei 100, Taiwan; ycheng@ntu.edu.tw; 8Institute of Environmental and Occupational Health Sciences, College of Public Health, National Taiwan University, Taipei 100, Taiwan; 9Department of Public Health, College of Medicine, National Cheng Kung University, Tainan 701, Taiwan; jdwang121@gmail.com; 10Department of Occupational and Environmental Medicine, National Cheng Kung University Hospital, College of Medicine, National Cheng Kung University, Tainan 701, Taiwan

**Keywords:** asbestos, gastric cancer, malignant mesothelioma, occupational cancer, standardized incidence ratio

## Abstract

Asbestos has been recognized as a human carcinogen associated with malignant mesothelioma, cancers of lung, larynx, and ovary. However, a putative association between gastric cancer and asbestos exposure remains controversial. In this study, we aimed to explore gastric cancer risk of workers potentially exposed to asbestos in Taiwan. The asbestos occupational cohort was established from 1950 to 2015 based on the Taiwan Labor Insurance Database, and Taiwan Environmental Protection Agency regulatory datasets, followed by the Taiwan Cancer Registry for the period 1980–2015. Standardized incidence ratios (SIRs) for cancer were computed for the whole cohort using reference rates of the general population, and also reference labor population. Compared with the general population, SIR of the asbestos occupational cohort for the gastric cancer increased both in males (1.05, 95% confidence interval (CI): 1.02–1.09) and females (1.10, 95% CI: 1.01–1.18). A total of 123 worksites were identified to have cases of malignant mesothelioma, where increased risk for gastric cancer was found with a relative risk of 1.76 (95% CI: 1.63–1.90). This 35-year retrospective cohort study of asbestos-exposed workers in Taiwan may provide support for an association between occupational exposure to asbestos and gastric cancer.

## 1. Introduction

Asbestos is termed “the magic mineral” with excellent properties in heat resistance, thermal and electrical insulation, tensile strength, and has gained extensive use in various industries, including architecture construction, ship building, heat insulation, friction materials, and textiles [1]. The global consumption of asbestos and related products reached the peak in the 1980s. However, many countries prohibit the use of asbestos product due to its adverse health effects, such as being a human carcinogen for decades [2,3]. The International Agency for Research on Cancer (IARC) has classified all types of asbestos as a group 1 human carcinogen. Many asbestos-related cancers have been identified, including malignant pleural mesothelioma (MPM), lung cancer, larynx cancer, and ovarian cancer [4]. The IARC reviews have provided up-to-date information on the cancer sites associated with each human carcinogen. Use of mechanistic data to identify carcinogens is increasing, and epidemiological research is identifying additional carcinogens and cancer sites or confirming carcinogenic potential under conditions of low exposure. A global analysis documented ecological association between ARD mortality rates in the 2000s and prior asbestos use in the 1960s, and also revealed a mean latency period up to 30–40 years [3]. As MPM is the signal tumor of asbestos exposure, many industrialized countries are expected to encounter MPM epidemics in the coming several decades [5], and our recent analysis of MPM incidence in Taiwan showed a latency period of 31 years, which is in accordance with the international trend [6]. There is a general lack of detailed exposure history and the long latency period associated with asbestos-related diseases (ARDs). Thus, it is usually difficult to establish a causal link to individual cancer clusters.

In our previous study, Lin et al. reported that males in the asbestos industry in Taiwan had about a three-fold increased risk of MPM for the period 1980–2009 [7]. Among them, those with more than 20 working years had the highest risk (about six-fold). However, even those employed less than 1 year could be associated with an elevated risk of MPM (2.6-fold), with a minimum latency period of 20 years.

In addition to inhalation exposure to asbestos fibers among workers, oral ingestion of the carcinogenic fibers may be associated with gastrointestinal (GI) cancers, especially gastric cancer. Gastric cancer is the fifth most common cancer worldwide and the third most deadly cancer [8]. The etiology of gastric cancer is multifactorial. Many studies have established causes of gastric cancer include alcohol drinking [9], foods preserved by salting, Helicobacter pylori infection, and tobacco smoking [10] as cofactors for its development. IARC summarized an insufficient evidence for gastric cancer associated with asbestos exposure. However, a case of gastric cancer associated with asbestosis was reported in Korea with estimated asbestos exposure of more than 38 fibers/mL-year [9].

Goodman et al. [11] conducted a meta-analysis for 69 asbestos-exposed occupational cohorts to reveal meta-standardized mortality ratio (SMR) of 92 (95% confidence interval: 77–110) for gastric cancer with latency of at least 10 years. They found not enough evidence of a significant association or dose–response effect. A significant exposure–response trend was seen between asbestos exposure in Chinese chrysotile miners and gastric cancer mortality by Lin et al. in 2014 [12]. Moreover, a recent meta-analysis of 40 mortality cohort studies showed an overall meta-SMR for gastric cancer of 1.15 (95% confidence interval 1.03–1.27) [13]. Peng et al. [14] applied a meta-analysis of 32 studies to show elevated risk of gastric cancer mortality among workers exposed to crocidolite, especially male miners, with an SMR of 1.19 (95% CI 1.06–1.34). In summary, the relationship between gastric cancer and asbestos exposure remains controversial.

Most of the previous studies with positive association between occupational exposure to asbestos and gastric cancer were meta-analyses in collection of adequate sample size. In this study, we extended the previous cohort size and the follow-up period, and established a retrospective cohort using of Taiwan Labor Insurance Databases, and Environmental Protection Agency (EPA) regulatory datasets to identify those who worked in asbestos-related industries between 1950 and 2015 and followed up with the Taiwan Cancer Registry (TCR) for the period 1980–2015 to identify incident cancer. The present study aimed to explore gastric cancer risk of workers potentially exposed to asbestos in Taiwan based on the retrospective cohort study.

## 2. Materials and Methods

### 2.1. Study Population Data Source

This research was approved by the Institutional Review Board of National Health Research Institutes (NHRI IRB number EC1070702-E).

The asbestos occupational cohort was established from 1950 to 2015 based on Taiwan Labor Insurance Databases, and Taiwan Environmental Protection Agency regulatory datasets, followed by the Taiwan Cancer Registry for the period 1980–2015. A total of 1,043,319 workers who were ever employed in the 631 asbestos-related factories between 1 March 1950 and 31 December 2015 were included in the cohort. Detailed information about the previous cohort for the period from 1950 to 2009 is provided elsewhere [7].

We obtained incident cases of cancer through linkages with the database of TCR [15] using encrypted personal identification numbers of these workers. All cancer diagnoses in the TCR were described using the International Classification of Diseases, Oncology version 3 (ICD-O-3) codes [16]. Data from the TCR were highly reliable for the diagnosis of cancer and considered accurate in analyzing incidence rate.

We selected malignant mesothelioma (C45), for which there is sufficient evidence of the association with asbestos in the International Agency for Research on Cancer (IARC) classification. In addition, gastric cancer (C16) was selected as asbestos-related cancer. The pathological diagnosis of malignant pleural mesothelioma (MPM) was coded according to cancer site of pleura (C45.0), and morphologic codes (fibrous sarcomatoid (9051/3), epithelioid (9052/3), and biphasic (9053/3)), based on the ICD-O-3.

### 2.2. Data Analysis

The National Institute for Occupational Safety and Health (NIOSH) Life Table Analysis System (LTAS) (website link: http://www.cdc.gov/niosh/ltas/ (accessed on 30 April 2021)) was used to examine standard incidence rate of gastric cancer (SIR), using the Taiwanese general population as a reference. SIR was calculated as the ratio of observed malignancies to the expected number of cases estimated using the TCR dataset (1980–2015) published by the Health Promotion Administration, Ministry of Health and Welfare. Confidence limits for risk measures were estimated based on a Poisson distribution for the observed outcome, with exact limits for outcomes with 10 or fewer occurrences.

For the incident case analyses, person years at risk (PYAR) began on the date of cohort inclusion (the earliest being 1 March 1950) and ended with the earliest index date or 31 December 2015. We defined the PYAR as the sum of the products of number of workers who worked for more than 20 years times the mean follow-up years of the corresponding factory [17]. Follow-up years were calculated from the year enrolled in the factories to the year (1) incident cancer occurred, (2) censored, or (3) the end of the study, whichever came earliest.

Estimated rate ratios (eRR) were computed for worksites where any case of cancer of interest was diagnosed compared with those with no case or the whole asbestos cohort by assuming Poisson distribution. The 95% CIs of the eRRs were estimated by Byar’s approximation of the Poisson method [18]; SAS version 9.4 (Cary, NC, USA) was used to analyze data.

## 3. Results

### 3.1. Standardized Incidence Ratio (SIR) of Gastric Cancer

Our study showed that there was a significantly increased SIR of gastric cancer (SIR = 1.06, *p* = 0.01) in the occupational cohort of workers who potentially experience asbestos exposure, by taking the general population of Taiwan as reference (Table 1). Compared with the general population, SIR of the asbestos occupational cohort for the gastric cancer increased both in males (1.05, 95% CI: 1.02–1.09) and females (1.10, 95% CI: 1.01–1.18). It is important to note that the total amount of male workers (*n* = 3088) developing gastric cancer was far more than that of female (*n* = 677). However, the value of SIR for different genders cannot be compared directly.

### 3.2. Exposure Classification Using Malignant Mesothelioma (MM)

Malignant mesothelioma (MM) was the signal tumor of asbestos exposure and could be used as a proxy indicator of definite asbestos exposure in the workplace. The diffuse MM was defined as pathologically confirmed cases of MM of pleura or peritoneum. We identified the worksites where diffuse MM was diagnosed during the follow-up period from 1980 to 2015. A total of 123 worksites were identified to have cases of diffuse MM. The industries having asbestos-exposed workers were divided into two subgroups: diagnosed as diffuse MM, and without the diagnosis of diffuse MM. Total person-year at risk was 17,557,645 and 6,969,593 among the two subgroups, respectively. Higher absolute incidence rates (2.08 × 10^−4^) can be found in the subgroup of industries with diffuse MM, compared to the industries without diffuse MM (1.18 × 10^−4^), which is shown in Table A1 (Appendix A). Moreover, an increased person-year at risk could be found among male workers (16,733,690) than female workers (7,753,548). Male workers were also having higher incidence rate of gastric cancer (2.19 × 10^−4^) than female workers (1.05 × 10^−4^) (Table A1 Appendix A).

Table 2 shows the estimated rate ratio of developing gastric cancer in asbestos-exposed workers. It is important to note that a higher eRR can be found in the workers contributing to the industries with diffuse MM, which further highlights the coincidence of asbestos-related MM and gastric cancer. By using diffuse MM as exposure classification, the industries with diffuse MM consistently presented significantly increased relative risks compared to industries without diffuse MM or the whole asbestos cohort.

### 3.3. Stratification by Latency and Employment Period

We defined latency as time from the first employment to diagnosis of cancer. In Table 3, we analyze different latencies of asbestos-exposed workers (1–9 years, 10–19 years, 20–29 years, and >30 years) by stratifying the workers according to their working durations registered on labor insurance (<1 year, 1–9 years, 10–19 years, and >20 years).

Table 3 reveals that the overall risk for male workers exposed to asbestos developing gastric cancer was 21% higher compared to the general worker population (SIR = 1.21, 95% CI 1.16–1.25). The high SIR values can be found in those who have employment periods of 1 to 9 years (1.26, 95% CI 1.19–1.32) and 10 to 19 years (1.29, 95% CI 1.19–1.40), respectively, which supports the increase of risk with longer employment period. For the latency of developing gastric cancer, significantly increased SIR was observed in the subgroup of latency of 10 to 19 years (1.33, 95% CI 1.24–1.43), and 20–29 years (1.21, 95% CI 1.14–1.29), and the highest SIR value was within the subgroup of less than 10 years (1.46, 95% CI 1.32–1.60).

Among the female workers experiencing asbestos exposure, the overall risk of developing gastric cancer was 16% higher when compared to the reference female workers’ population (1.16, 95% CI 1.08–1.25). Similar to the results of male workers, the highest SIR value of latency of gastric cancer was less than 10 years (SIR = 1.30, 95% CI 1.11–1.51). Besides, the significant increased SIR can also be found in the subgroup of latency of 10 to 19 years (SIR = 1.23, 95% CI 1.07–1.41), as well as the subgroup of latency between 20 to 29 years (SIR = 1.17, 95% CI 1.01–1.34).

## 4. Discussion

This study, based on a large retrospective cohort with a follow-up of more than 10 years, supports a positive association between cumulative exposure to asbestos and the incidence of stomach cancer. The asbestos occupational cohort was established from 1950 to 2015 based on Taiwan Labor Insurance Databases, and Taiwan Environmental Protection Agency regulatory datasets, followed by the TCR for the period 1980–2015, which found that, compared with the general population, SIR of the asbestos occupational cohort for the stomach cancer increased both in males (1.05, 95% confidence interval: 1.02–1.09) and females (1.10, 95% CI: 1.01–1.18). Using MM as a proxy indicator of definite asbestos exposure in exposure classification, we identified 123 worksites to have cases of MM, where increased risk for gastric cancer was found with an estimated rate ratio of 1.76 (95% CI: 1.63–1.90). As stratified by employment periods, increased risk of gastric cancer was observed with longer employment period. Our findings may provide support for an association between occupational exposure to asbestos and gastric cancer.

Nowadays, most studies on causal agents of gastric cancer focus on the diet and lifestyle factors. However, several cohort studies and case–control studies have reported occupational or environmental factors associated with gastric cancer, such as asbestos, mineral dusts, and N-nitroso compound [19]. The causal association between asbestos exposure and gastrointestinal (GI) cancers is still under debate, and few studies have provided only limited evidence.

Early in 1979, Selikoff et al. had reported that there was a three-fold increased risk of GI cancers among 632 building trades insulation workers with asbestos exposure [20]. Lin et al. had proposed a significant association between asbestos dust exposure and mortality from gastric cancer, with SMR of 2.39 (95% CI 1.02 to 5.60) and 6.49 (95% CI 2.77 to 15.20) in a cohort of 1539 asbestos miners [11]. Besides, the study by Pang et al. [21] involved Chinese chrysotile factory workers with 7.9-fold excess risk of five males developing gastric cancer cases and a 4.4-fold excess when women were included. Significantly elevated SMR (1.24) was found in a cohort study of 26 workers exposed to chrysotile alone by Li et al. [22]. A prospective cohort study in Netherland proposed that there was a significant relationship between long-time exposure of asbestos and gastric cancer, as well as gastric noncardiac adenocarcinoma [23]. Moreover, a meta-analysis by Fortunato and Rushton [13] disclosed the statistically significant difference between occupational asbestos exposure and gastric cancer, with meta-SMR = 1.15 (95% CI 1.03–1.27). A dramatic increased trend of male gastric cancer, with SMR = 1.46 (95% CI 1.22–1.77), can be found in the sub-stratification group of lung cancer having SMR > 2, whereas female with SMR = 1.02 (95% CI 0.69–1.77). A more prominent magnitude of relative risk of male gastric cancer, with SMR = 1.91 (95% CI 1.03–3.56), can be noted in the subgroup sample of occupational cohorts in China and Russia. Barbiero et al. reported a borderline increased incidence found for gastric cancer (SIR = 1.53, 95% CI 0.96–2.31) in an Italian cohort enrolled in a public health surveillance program [24]. A case of gastric cancer was recognized as work-related disease in a 57-year-old Korean man who had continuous exposure to high levels of asbestos for 40 years [25]. However, he had a history of smoking and drinking, thus the recognition of occupational cancer for compensation needs to be judged carefully with consideration of an individual’s nonoccupational personal risk factors and intensity of occupational exposure. Our findings of modest association (relative risk less than 2) may corroborate some positive relationship between asbestos and gastric cancer in international literature.

Relevant evidence shows that the most likely route between nonoccupational and environmental asbestos exposure related to GI cancers is ingestion, such as the drinking water contaminated by chrysotile, through GI tract. A recent systemic review study highlighted a positive correlation between ingested asbestos and gastric cancer risk based on a Norwegian study [26]. This retrospective study in Norway reported that in the subgroup of workers who had definite exposure to asbestos, SIR of gastric cancer was up to 2.5 (CI: 0.9–5.5) [27]. This finding may corroborate that the ingested asbestos can attack the gastrointestinal mucosa and interfere with the regulation of steady-state DNA synthesis in the GI tract. Reduced DNA synthesis can downregulate the cancer cell proliferation, resulting in neoplastic transformation. Another hypothesis in cellular immunity pathway showed significantly lower values of phytohemagglutinin-induced proliferative and cytotoxicity in the patients exposed to asbestos. In a nutshell, this evidence points out that all forms of asbestos exposure could increase the risk of gastric cancer modestly. Further studies are needed to investigate the molecular pathophysiology of carcinogenesis of gastric cancer associated with asbestos.

According to WHO estimates, asbestos exposure would account for more than 100,000 deaths per year in the world, making it one of the top public health issues [28]. Some countries in Asia and the Middle East continue to use asbestos and consume asbestos-related products [29], despite more than 60 countries in the world having a total ban on asbestos [30]. In Taiwan, asbestos has been listed as a toxic chemical by the Taiwan EPA since 1989, with phased prohibitions on asbestos use over several years subsequently, and there have been no permits issued for the manufacture of asbestos products since 1 January 2018 [31]. However, asbestos fibers were used in producing brake linings until 2018 and may be found in fire-resistant materials in building construction and old factories. Thus, the risks of environmental asbestos exposure still exist, and we should take the right action to prevent asbestos exposure from use of asbestos-related products and application as soon as possible. Furthermore, potential challenges of a third wave of ARDs [32] that may result from exposure during demolition or replacement operations at sites with “asbestos in place” [33] should be well-considered and prepared for.

This study has the following limitations that should be addressed. First, we did not have exposure data on ambient air levels of asbestos fibers, and heterogeneity of exposure to asbestos within the asbestos occupational cohort may be substantial. Applying a job-exposure matrix for exposure assessment was impossible in our study. Malignant mesothelioma diagnosed in the worksites was used for exposure classification in our study design, and our findings of consistently increased estimated rate ratios may corroborate the significant association. Second, the risk factors associated with gastric cancer, such as alcohol drinking, cigarette smoking, chronic infection of Helicobacter pylori, and diet habits, were not available in the administrative databases used in this study. We were unable to control the potential confounding factors in the causality of gastric cancer. Third, the first employment in the potentially asbestos-exposed industry was assumed the start point of employment period or latency period, which may overestimate the true length of employment or latency. Such overestimation might affect assessment of the minimal latency from the first exposure to cancer diagnosis. Readers need to be cautious in interpretation of the study findings given these study limitations.

In summary, our results demonstrate that exposure to asbestos was associated with an increased risk of gastric cancer. Future research may be considered to examine any exposure dose–response relationship using cumulative exposure indices to asbestos fibers or taking malignant mesothelioma as a surrogate marker of asbestos-related diseases. Future studies may collect information of potential confounding factors, such as alcohol drinking, cigarette smoking, or Helicobacter pylori infection, and then control these factors for causal inference.

## 5. Conclusions

We found a significantly elevated standardized incidence ratio of gastric cancer in asbestos-exposed workers in a 35-year cohort study of asbestos-exposed workers in Taiwan. Our findings may provide support for an association between occupational exposure to asbestos and gastric cancer. Further studies are needed to verify causality through examining the dose–response relationship, considering occurrence of malignant mesothelioma and controlling confounding factors.

## Figures and Tables

**Table 1 ijerph-18-07521-t001:** Standardized incidence ratio (SIR) of gastric cancer among the occupational cohort of worker exposed to asbestos, using the general population of Taiwan as reference.

Cancer Site	ICD-10 Code	Observed No.	Expected No.	SIR (95% CI)
Gastric cancer				
Male	C16	3088	2928.78	1.05 ** (1.02–1.09)
Female	C16	677	618.03	1.10 * (1.01–1.18)
Total		3765	3546.81	1.06 ** (1.03–1.10)

* *p* < 0.05; ** *p* < 0.01. 95% confidence interval (CI). International Classification of Diseases (ICD)-10 code for gastric cancer is C16.

**Table 2 ijerph-18-07521-t002:** Estimated rate ratio (eRR) of gastric cancer among asbestos-exposed workers in industries with diffuse malignant mesothelioma (MM).

Total	eRR of Gastric Cancer
Industries with diffuse MM compared to the industries without diffuse MM	1.76 (95% CI: 1.63 to 1.90)
Industries with diffuse MM compared to the whole asbestos cohort	1.14 (95% CI: 1.09 to 1.19)
Male	
Industries with diffuse MM compared to the industries without diffuse MM	1.54 (95% CI: 1.41 to 1.67)
Industries with diffuse MM compared to the whole asbestos cohort	1.10 (95% CI: 1.04 to 1.15)
Female	
Industries with diffuse MM compared to the industries without diffuse MM	2.17 (95% CI: 1.83 to 2.58)
Industries with diffuse MM compared to the whole asbestos cohort	1.23 (95% CI: 1.11 to 1.37)

95% confidence interval (CI).

**Table 3 ijerph-18-07521-t003:** Standardized incidence ratio (SIR) of gastric cancer among Taiwan asbestos-exposed workers, stratified by the employment period registered on labor insurance, using controlled generation of workers as reference.

Employment Period	Latency of Gastric Cancer (ICD-C16) among Taiwan Asbestos-Exposed Workers				
	<10 Years	10–19 Years	20–29 Years	≥30 Years	Subtotal
Males					
<1 year	1.45 (1.21–1.71) **	1.23 (1.05–1.42) **	1.06 (0.92–1.22)	0.89 (0.77–1.02)	1.09 (1.02–1.18) *
1–9 years	1.46 (1.30, 1.63) **	1.32 (1.19–1.46) **	1.30 (1.18–1.42) **	1.04 (0.94–1.15)	1.26 (1.19–1.32) **
10–19 years	0.00 (0.00–51,565.74)	1.44 (1.27–1.63) **	1.24 (1.07–1.42) **	1.16 (0.98–1.36)	1.29 (1.19–1.40) **
≥20 years	-	0.00 (0.00–5878.35)	1.16 (0.95–1.40)	1.00 (0.82–1.20)	1.07 (0.93–1.22)
subtotal	1.46 (1.32–1.60) **	1.33 (1.24–1.43) **	1.21 (1.14–1.29) **	1.01 (0.94–1.08)	1.21 (1.16–1.25) **
Females					
<1 year	1.22 (0.89–1.64)	0.96 (0.68–1.32)	1.29 (0.98–1.67)	0.97 (0.68–1.33)	1.11 (0.95–1.29)
1–9 years	1.33 (1.11–1.59) **	1.48 (1.20–1.81) **	1.02 (0.78–1.32)	0.73 (0.50–1.03)	1.19 (1.06–1.33) **
10–19 years	0.00 (0.00–1,647,952.97)	1.16 (0.92–1.44)	1.34 (0.98–1.78)	0.99 (0.54–1.66)	1.19 (1.00–1.40) *
≥20 years	-	0.00 (0.00–670,749.35)	1.09 (0.76–1.52)	1.04 (0.58–1.72)	1.08 (0.80–1.42)
subtotal	1.30 (1.11–1.51) **	1.23 (1.07–1.41) **	1.17 (1.01–1.34) *	0.89 (0.72–1.08)	1.16 (1.08–1.25) **

* *p* < 0.05; ** *p* < 0.01.

## Data Availability

No additional data are available.

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
