# Peer review of "Increased Risk of Gastric Cancer in Asbestos-Exposed Workers: A Retrospective Cohort Study Based on Taiwan Cancer Registry 1980–2015"

_ijerph, 2021, doi:10.3390/ijerph18147521_

Round 1

Reviewer 1 Report

Very good and interesting paper. I thought that the paper was sound and it included critical information on the Asbestos in Taiwan. Here are my suggestions for change:

Abstract: Usage of acronyms is too frequent and this section needs to be rewritten.

Introductions: I noticed that introduction did not include much detail on study objectives, research questions, and hypotheses, which needs to be addressed. The rational for conducting the study needs to be clearer.

Methods: It would have been nice to conduct a Joinpoint analysis (https://doi.org/10.1016/j.heliyon.2019.e02515), possibly using this software (https://surveillance.cancer.gov/joinpoint/). I also would have included a survival analyses approach, using a life table.

Results: Tables need to stand alone and inserted into the manuscript.  I had a hard time rationalizing the table shells of data, and captions needed more context, including information on what the acronyms and letters signify. Joinpoint regression analysis should be included as Table 4/Figure 1. I also recommend presenting a life-table analysis as Figure 2.

Discussion: I also noticed that there was little attention to limitations, recommendations, and significance of study, with each being a distinct subsection (i.e., limitations outlines methodological and practical issues with the paper; recommendation outlines areas for future research; significance outlines areas of change). I also would have like to include more information relating your study to international literature.

Conclusions: This section is too short. Paragraphs need to include at least three sentences. This section needs be rewritten, this reads more of a recommendations section and less of a paragraph to conclude a paper, which needs to be supported by the results.

References:  I also found that the reference section uses an inconsistent format, which could have been corrected by using a reference management system, such as open-source Zotero (https://www.zotero.org/), which provides an add-in to Microsoft Word and modern browsers. It would have been nice if you included DOIs or hyperlinks for each reference.

Reviewer 2 Report

Dear Editor,

Thanks for the opportunity to review this manuscript titled “Increased risk of gastric cancer in asbestos-exposed workers: A retrospective cohort study based on Taiwan Cancer Registry 1980-2015” proposed by Fang and co-workers.

The topic is interesting, but the presentation needs great improvement, especially regarding the English language. In addition, there is really no discussion, which should be rewritten. My specific comments are found below.

Abstract

The section is informative but requires substantial editing for language improvement. These are just examples (the authors should correct the whole section or seek help from a native English speaking scientist):

  1. Standardized incidence ratio (SIR) for cancer were ...
  2. This 35-year cohort study of asbestos-exposed workers in Taiwan may be positively associated with gastric cancer

Introduction

"Asbestos is a magic mineral..." should be "Asbestos is also termed "the magic mineral"..."

The logic in this section should be improved. "The International Agency for Research on Cancer has classified all types of asbestos as a group 1 human carcinogen" and "Asbestos has been recognized as a...[3]" should be close to each other. In addition, there is no transition before "Gastric cancer is the fifth most common..."

"However, conclusions of studies have not reached a consensus" should be rephrased

"Lin et al. reported that ..." should be placed before information about gastric cancer

The authors could include a clear study hypothesis and/or aim.

The section also requires substantial editing for language improvement

Materials and Methods

"... (C18), and rectal cancer (C19–C20), and renal cancer (C64)..." should be corrected for punctuation.

" ...of malignant pleural mesothelioma (MPM) is coded...": "is" should be "was".

The section also requires substantial editing for language improvement

Results

The section should be structured with different paragraphs and appropriate titles

past tense should be used throughout

"The industries having asbestos-exposed are divided into two subgroups: diagnosed as diffuse MM and without the diagnosis of diffuse MM" is unclear

The section also requires substantial editing for language improvemen

Table 2 could be deleted, and the information added to the main text.

Discussion

This section looks like a second Introduction. It should be rewritten.

While doing so, the authors could take into consideration the following:

  1. Start with a brief summary of principal findings
  2. Present the findings in light of previously published data
  3. Add some study limitations
  4. End with a conclusion (the present one is incomplete and unclear).

Reviewer 3 Report

This manuscript is an original article that retrospectively investigated the association between gastric cancer and asbestos-exposure based on databases in Taiwan. The authors showed standard incidence rate of gastric cancer was increased in workers with asbestos-exposure. Furthermore, they showed cases with malignant mesothelioma had increased risk for gastric cancer. This study was conducted well, and the methods are appropriate. The data are presented clearly. The results will be of interest to clinicians in the field. However, the following major and minor issues require clarification: Major 1. Previous studies have already shown the relationship between gastric cancer and asbestos-exposure as the authors described in the discussion section. Therefore, it’s difficult to find novelty and originality in this study. The authors should emphasize them in the discussion, conclusion and abstract. Please expand discussion in contradistinction to the results in previous studies. Minor 1. Please explain the abbreviations such as “MM” and “SMR”. 2. Did the authors use the data other than gastric cancer and malignant methothelioma? If they were not used, please delete them from Data source. 3. I recommend the authors explain the results regarding Table 2 in more detail. 4. (P4L160) The SIR value in those who have employment periods of 1 to 9 years did not match with that in Table 3. 5. (P4L162-163) This sentence (which can be partially attributed to -) should be described in discussion section as it is not the result. 6. (Table 3) Please replace “sutotal” with “subtotal”.

Round 2

Reviewer 1 Report

I felt like the author's addressed most of my concerns with the manuscript and it ended up significantly improving the quality of the article. Here are some suggestions for improvement:

  1. Are there any specific aims, research questions, or hypotheses should should be added in Line 121.
  2. Do you have any answers to the specific aims, research questions, or hypotheses, which should be addressed in discussion or conclusion sections.
  3. Do you have any strengths or areas of future research that could be explored in future studies?
  4. I suggest adding the supplementary table as Appendix A to avoid having people look at multiple documents.
  5. Make sure that captions appear on the same page as table or figures.
  6. Paper includes numbers without thousands separator and scientific notation, which needs to be avoided (lines 190-196) and converted into standard notation.
  7. Tables need to be stand alone and needs to include more footnotes. I am unclear what C16 in Table 1, 95CI% in table 2 are without footnotes or clearer mentions in the table shells.
  8. References need to include more recent sources (2017 or later). I strongly discourage older sources because they are not up-to-date and information might not be valid for this study.

Good luck with the revisions.

Reviewer 2 Report

The authors have addressed the quasi-totality of my comments. However, the whole manuscript should be proofread by a native English speaking scientist before publication.

Reviewer 3 Report

Thank you for revising the manuscript according to my comments. The revised manuscript is much improved; however, the following minor issues require clarification:

  1. The authors didn’t use selected data of lung cancer (C33–C34), laryngeal cancer (C32), ovarian cancer (C56), pharyngeal cancer (C10–C13), esophageal cancer (C15), colon cancer (C18), and rectal cancer (C19–C20), and renal cancer (C64) in this study. Then, the authors should delete them from Study Population Data Source.

  1. I recommend that the authors should delete the last sentence in the conclusion as it seems overlapped.
